# Fairness for Workers Who Pull the Arms: An Index Based Policy for Allocation of Restless Bandit Tasks

## Abstract

Motivated by applications such as machine repair, project monitoring, and anti-poaching patrol scheduling, we study intervention planning of stochastic processes under resource constraints. This planning problem has previously been modeled as restless multi-armed bandits (RMAB), where each arm is an intervention-dependent Markov Decision Process. However, the existing literature assumes all intervention resources belong to a single uniform pool, limiting their applicability to real-world settings where interventions are carried out by a set of workers, each with their own costs, budgets, and intervention effects. In this work, we consider a novel RMAB setting, called multi-worker restless bandits (MWRMAB) with heterogeneous workers. The goal is to plan an intervention schedule that maximizes the expected reward while satisfying budget constraints on each worker as well as fairness in terms of the load assigned to each worker. Our contributions are two-fold: (1) we provide a multi-worker extension of the Whittle index to tackle heterogeneous costs and per-worker budget and (2) we develop an index-based scheduling policy to achieve fairness. Further, we evaluate our method on various cost structures and show that our method significantly outperforms other baselines in terms of fairness without sacrificing much in reward accumulated.

## 1 Introduction

Restless multi-armed bandits (RMABs) Whittle [1988] have been used for sequential planning, where a planner allocates a limited set of $M$ *intervention resources* across $N$ *independent heterogeneous arms* (Markov Decision processes) at each time step in order to maximize the long-term expected reward. The term *restless* denotes that the arms undergo state-transitions even when they are not acted upon (with a different probability than when they are acted upon). RMABs have been receiving increasing attention across a wide range of applications such as maintenance [Abbou and Makis, 2019], recommendation systems Meshram *et al.* [2015], anti-poaching patrolling [Qian *et al.*, 2016b], and adherence monitoring [Akbarzadeh and Mahajan, 2019; Mate *et al.*, 2020]. Although, *rangers* in anti-poaching, *healthcare workers* in health intervention planning, and *supervisors* in machine maintenance are all commonly cited examples of human workforce used as intervention resources, the literature has so far ignored one key reality that the human workforce is heterogeneous—each worker has their own workload constraints and needs to commit a dedicated time duration for intervening on an arm. Thus, it is critical to restrict intervention workload for each worker and balance the workload across them, while also ensuring high effectiveness (reward) of the planning policy.

RMAB literature does not consider this heterogeneity and mostly focuses on selecting best arms assuming that all intervention resources (workers) are interchangeable, i.e., as from a single pool (homogeneous). However, planning with human workforce requires more expressiveness in the model, including heterogeneity in costs and intervention effects, worker-specific load constraints, and

Submitted to 36th Conference on Neural Information Processing Systems (NeurIPS 2022). Do not distribute.

balanced work allocation. One concrete example is *anti-poaching intervention planning* Qian *et al.* [2016a] with $N$ areas in a national park where timely interventions (patrols) are required to detect as many snares as possible across all the areas. These interventions are carried out by a small set of $M$ ranger. The problem of selecting a subset of areas at each time step (say, daily) has been modeled as an RMAB problem. However, each ranger may incur heterogeneous cost (e.g., distance travelled, when assigned to intervene on a particular area) and the total cost incurred by any ranger (e.g., *total distance traveled*) must not exceed a given budget. Additionally, it is important to ensure that tasks are allocated fairly across rangers so that, for e.g., some rangers are not required to walk far greater distances than others. Adding this level of expressiveness to existing RMAB models is non-trivial.

To address this, we introduce the *multi-worker restless multi-armed bandits* (MWRMAB) problem. Since MWRMABs are more general than the classical RMABs, they are at least PSPACE hard to solve optimally [Papadimitriou and Tsitsiklis, 1994]. RMABs with $k$-state arms require solving a combined MDP with $k^N$ states and $|M + 1|^N$ actions constrained by a budget, and thus suffers from the curse of dimensionality. A typical approach is to compute Whittle indices [Whittle, 1988] for each arm and choose $M$ arms with highest index values—an asymptotically optimal solution under the technical condition *indexability* [Weber and Weiss, 1990]. However, this approach is limited to instances a single type of intervention resource incurring one unit cost upon intervention. A few papers on RMABs [Glazebrook *et al.*, 2011; Meshram and Kaza, 2020] study multiple interventions and non-unitary costs but assumes one global budget (instead of per-worker budget). Existing solutions aim at maximizing reward by selecting arms with highest index values that may not guarantee fairness towards the workers who are in charge of providing interventions.

To the best of our knowledge, we are the first to introduce and formalize the multi-worker restless multi-armed bandit (MWRMAB) problem and a related worker-centric fairness constraint. We develop a novel framework for solving the MWRMAB problem. Further, we empirically evaluate our algorithm to show that it is fair and scalable across a range of experimental settings.

## 2  Related Work

**Multi-Action RMABs and Weakly Coupled MDPs** Glazebrook *et al.* [2011] develop closed-form solutions for multi-action RMABs using Lagrangian relaxation. Meshram and Kaza [2020] build simulation-based policies that rely on monte-carlo estimation of state-action values. However, critically, these approaches rely on actions being constrained by a single budget, failing to capture the heterogeneity of workforce. On the other hand, weakly coupled MDPs (WCMDPs) Hawkins [2003] allow for such multiple budget constraints; this is the baseline we compare against. Other theoretical works Adelman and Mersereau [2008]; Gocgun and Ghate [2012] have developed solutions in terms of the reward accumulated, but may not scale well with increasing problem size. These papers do not consider fairness, a crucial component of MWRMABs, which our algorithm addresses.

**Fairness** in stochastic and contextual multi-armed bandits (MABs) [Patil *et al.*, 2020; Joseph *et al.*, 2016; Chen *et al.*, 2020] has been receiving significant attention. However, fairness in RMABs has been less explored. Recent work by Herlihy *et al.* [2021] considered quota-based fairness of RMAB arms assuming that arms correspond to human beneficiaries (for example, patients). However, in our work, we consider an orthogonal problem of satisfying the fairness among intervention resources (workers) instead of arms (tasks).

**Fair allocation** of discrete items among a set of agents has been a well studied topic [Brandt *et al.*, 2016]. Fairness notions such as envy-freeness up to one item [Budish, 2011] and their budgeted settings [Wu *et al.*, 2021; Biswas and Barman, 2018] align with the fairness notion we consider. However, these papers do not consider non-stationary (MDP) items. Moreover, these papers assume that each agent has a value for every item; both fairness and efficiency are defined with respect to this valuation. In contrast, in MWRMAB, efficiency is defined based on reward accumulated and fairness and budget feasibility are defined based on the cost incurred.

## 3  The Model

There are $M$ workers for providing interventions on $N$ independent arms that follow Markov Decision Processes (MDPs). Each MDP $i \in [N]$ is a tuple $\langle S_i, A_i, C_i, P_i, R_i \rangle$, where $S_i$ is a finite set of states. We represent each worker as an action, along with an additional action called *no-intervention*. Thus,

89 action set is $A_i \subseteq [M] \cup \{0\}$. $C_i$ is a vector of costs $c_{ij}$ incurred when an action $j \in [A_i]$ is taken on
90 an arm $i \in [N]$, and $c_{ij} = 0$ when $j = 0$. $P_{ij}^{ss'}$ is the probability of transitioning from state $s$ to state
91 $s'$ when arm $i$ is allocated to worker $j$. $R_i(s)$ is the reward obtained in state $s \in S_i$.

92 The goal (Eq. 1) is to allocate a subset of arms to each worker such that the expected reward is
93 maximized while ensuring that each worker incurs a cost of at most a fixed value $B$. Additionally,
94 the disparity in the costs incurred between any pair of workers does not exceed a *fairness threshold $\epsilon$*
95 at a given time step. Let us denote a policy $\pi : \times_i S_i \mapsto \times_i A_i$ that maps the current state profile of
96 arms to an action profile. $x_{ij}^\pi(s) \in \{0, 1\}$ indicates whether worker $j$ intervenes on arm $i$ at state $s$
97 under policy $\pi$. The total cost incurred by $j$ at a time step $t$ is given by $\overline{C}_j^\pi(t) := \sum_{i \in N} c_{ij} x_{ij}^\pi(s_i(t))$,
98 where $s_i(t)$ is the current state. $\epsilon \geq c^m := \max_{ij} c_{ij}$ ensures feasibility of the fairness constraints.

$$
\begin{aligned}
\max_\pi \limsup_{T \to \infty} \frac{1}{T} \sum_{i \in [N]} \mathbb{E} & \left[ \sum_{t=1}^T R_i(s_i(t)) \, x_{ij}^\pi(s_i(t)) \right] \\
\text{s.t.} \sum_{i \in N} x_{ij}^\pi(s_i(t)) \, c_{ij} & \leq B, && \forall \, j \in [M], \, \forall \, t \in \{1, 2, \ldots\} \\
\sum_{j \in A_i} x_{ij}^\pi(s_i(t)) & = 1, && \forall \, i \in [N], \, \forall \, t \in \{1, 2, \ldots\} \\
\max_j \overline{C}_j^\pi(t) - \min_j \overline{C}_j^\pi(t) & \leq \epsilon, && \forall \, t \in \{1, 2, \ldots\} \\
x_{ij}^\pi(s_i(t)) & \in \{0, 1\}, && \forall i, \, \forall j, \, \forall t.
\end{aligned}
\tag{1}
$$

99 When $M = 1$ and $c_{i1} = 1$, Problem (1) becomes classical RMAB problem (with two actions,
100 *active* and *passive*) that can be solved via Whittle Index method [Whittle, 1988] by considering a
101 time-averaged relaxed version of the budget constraint and then decomposing the problem into $N$
102 subproblems—each subproblem finds a **charge** $\lambda_i(s)$ on active action that makes passive action as
103 valuable as the active action at state $s$. It then selects top $B$ arms according to $\lambda_i$ values at their
104 current states. However, the challenges involved in solving a general MWRMAB (Eq. 1) are (i) index
105 computation becomes non-trivial with $M > 1$ workers and (ii) selecting top arms based on indices
106 may not satisfy fairness. To tackle these challenges, we propose a framework in the next section.

## 4 Methodology

107

108 **Step 1**: Decompose the combinatorial MWRMAB problem to $N \times M$ subproblems, and compute
109 Whittle indices $\lambda_{ij}^\star$ for each subproblem. We tackle this in Sec. 4.1. This step assumes that, for each
110 arm $i$, MDPs corresponding to any pair of workers are mutually independent. However, the expected
111 value of each arm may depend on interventions taken by multiple workers at different timesteps.
112 **Step 2**: Adjust the decoupled indices $\lambda_{ij}^*$ to create $\lambda_{ij}^{adj,*}$, detailed in Sec. 4.2.
113 **Step 3**: The adjusted indices are used for allocating the arms to workers while ensuring **fairness** and
114 **per-timestep budget feasibility** among workers, detailed in Sec. 4.3.

### 4.1 Identifying subproblem structure

115

116 To arrive at a solution strategy, we relax the per-timestep budget constraints of Eq. 1 to time-
117 averaged constraints, as follows: $\frac{1}{T} \sum_{i \in [N]} \mathbb{E} \sum_{t=1}^T x_{ij}^\pi(s_i(t)) c_{ij} \leq B, \, \forall j \in [M]$. The optimization
118 problem (1) can be rewritten as:

$$
\begin{aligned}
\min_{\{\lambda_j \geq 0\}} \max_\pi \limsup_{T \to \infty} \frac{1}{T} \sum_{i \in [N]} \mathbb{E} & \left[ \sum_{t=1}^T \left( R_i(s_i(t)) x_{ij}^\pi(s_i(t)) + \sum_{j \in [M]} \lambda_j (B - c_{ij} x_{ij}^\pi(s_i(t))) \right) \right] \\
\text{s.t.} \sum_{j \in A_i} x_{ij}^\pi(s_i(t)) & = 1, && \forall \, i \in [N], \, t \in \{1, 2, \ldots\} \\
\max_j \overline{C}_j^\pi(t) - \min_j \overline{C}_j^\pi(t) & \leq \epsilon, && \forall \, t \in \{1, 2, \ldots\} \\
x_{ij}^\pi(s_i(t)) & \in \{0, 1\}, && \forall i, \, \forall j, \, \forall t
\end{aligned}
\tag{2}
$$

Here, $\lambda_j$s are Lagrangian multipliers corresponding to each relaxed budget constraint $j \in [M]$. Furthermore, as mentioned in Glazebrook *et al.* [2011], if an arm $i$ is *indexable*, then the optimization objective (2) can be decomposed into $N$ independent subproblems, and separate index functions can be defined for each arm $i$. Leveraging this, we decompose our problem to $N \times M$ subproblems, each finding the minimum $\lambda_{ij}$ that maximizes the following:

$$\limsup_{T \to \infty} \frac{1}{T} \mathbb{E}\left[\sum_{t=1}^{T} \left(R_i(s_i(t)) - \lambda_{ij} c_{ij}\right) x_{ij}^{\pi}(s_i(t))\right] \tag{3}$$

Note that, the maximization subproblem (3) does not have the term $\lambda_{ij} B$ since the term does not depend on the decision $x_{ij}^{\pi}(s_i(t))$. Considering a 2-action MDP with action space $\mathcal{A}_{ij} = \{0, j\}$ for an arm-worker pair, the maximization problem (3) can be solved by dynamic programming methods using Bellman's equations for each state to decide whether to take an active action ($x_{ij}(s) = 1$) when the arm is currently at state $s$:

$$V_{i,j}^{t}(s, \lambda_{ij}, x_{ij}(t)) = \begin{cases} R_i(s) - \lambda_{ij} c_{ij} + \sum\limits_{s' \in S_i} P_{ss'}^{ij} V_{i,j}^{t+1}(s', \lambda_{ij}), & \text{if } x_{ij}(t) = 1 \\ R_i(s) + \sum\limits_{s' \in S_i} P_{ss'}^{i0} V_{i,j}^{t+1}(s', \lambda_{ij}), & \text{if } x_{ij}(t) = 0 \end{cases} \tag{4}$$

$$\lambda_{ij}^{\star}(s) = \arg\min\{\lambda : V_{i,j}^{t}(s, \lambda, j) == V_{i,j}^{t}(s, \lambda, 0)\} \tag{5}$$

We compute the Whittle indices $\lambda_{ij}^{\star}$ (Eq. 5) [Qian *et al.*, 2016b] (the algorithm is in Appendix A).

Additionally, we establish that the Whittle indices of multiple workers are related when the costs and transition probabilities possess certain characteristics, enabling simplification of Whittle Index computation for multiple workers when there are certain structures in the MWRMAB problem.

**Theorem 1.** *For an arm $i$, and a pair of workers $j$ and $j'$ such that $c_{ij} \neq c_{ij'}$ and $P_{ss'}^{ij} = P_{ss'}^{ij'}$ for every $s, s' \in \mathcal{S}_i$, then their Whittle Indices are inversely proportional to their costs.*

$$\frac{\lambda_{ij}^{\star}(s)}{\lambda_{ij'}^{\star}(s)} = \frac{c_{ij'}}{c_{ij}} \text{ for each state } s \in \mathcal{S}_i$$

*Proof.* Let us consider an arm $i$ and a pair of workers $j$ and $j'$ such that $P_{ss'}^{ij} = P_{ss'}^{ij'}$. By definition of Whittle Index $\lambda_j(s)$ for a worker $j$, it is the minimum value at a state $s$ such that,

$$V_{ij}(s, \lambda_j(s), j) - V_{ij}(s, \lambda_j(s), 0) = 0 \tag{6}$$

Eq. 6 can be rewritten by expanding the value functions as:

$$R_i(s) - \lambda_j(s) c_{ij} + \sum_{s' \in \mathcal{S}_i} P_{ss'}^{ij} V_i(s', \lambda_j(s)) - R_i(s) + \sum_{s' \in \mathcal{S}_i} P_{ss'}^{i0} V_i(s', \lambda_j(s)) = 0$$

$$\implies -\lambda_j(s) c_{ij} + \sum_{s' \in \mathcal{S}_i} P_{ss'}^{ij} V_i(s', \lambda_j(s)) - \sum_{s' \in \mathcal{S}_i} P_{ss'}^{i0} V_i(s', \lambda_j(s)) = 0 \tag{7}$$

where, $V_i(s', \lambda_j(s')) = \max\limits_{a = \{0, j\}} R_i(s) - a\lambda_j(s) c_{ij} + \mathbb{E}_{s''}[V_i(s'', \lambda(s))]$.

Next, we substitute all $\lambda_j(s)$ terms by $\frac{x}{c_{ij}}$. After substitution, Eq. 7 is a function of $x$ only, i.e., no $\lambda(s)$ or $c_{ij}$ terms remain after substitution. We can rewrite Eq. 7 as:

$$-x + \sum_{s' \in \mathcal{S}_i} P_{ss'}^{ij} V_i(s', x) - \sum_{s' \in \mathcal{S}_i} P_{ss'}^{i0} V_i(s', x) = 0 \tag{8}$$

Note that $x^*$ that minimizes Eq. 8 corresponds to $\lambda_j(s) c_{ij}$ for any $j$, where $\lambda_j(s)$ is the Whittle index for worker $j$. Therefore, for any two workers $j$ and $j'$ with corresponding Whittle Indices as $\lambda_j(s)$ and $\lambda_{j'}(s)$, we obtain $\lambda_j(s) c_{ij} = \lambda_{j'}(s) c_{ij'}$ whenever $P_{ss'}^{ij} = P_{ss'}^{ij'}$. This completes the proof. □

Theorem 1 also implies that, when the costs and effectiveness of two workers are equal, then their Whittle indices are also equal, stated formally in Corollary 1.

**Corollary 1.** *For an arm $i$, and a pair of workers $j$ and $j'$ such that $c_{ij} = c_{ij'}$ and $P_{ss'}^{ij} = P_{ss'}^{ij'}$ for every $s, s' \in \mathcal{S}_i$, then their Whittle Indices are the same.*

$$\lambda_{ij}^{\star}(s) = \lambda_{ij'}^{\star}(s) \text{ for each state } s \in \mathcal{S}_i.$$

## 4.2 Adjusting for interaction effects

The indices obtained using Alg. 3 are not indicative of the true long-term value of taking that action in the MWRMAB problem. This is because, for a given arm, the value of an intervention by worker $j$ in general depends on interventions by other workers $j'$ at different timesteps.

Consider a 2-worker MWRMAB corresponding to an anti-poaching patrol planning problem, where each worker is a type of "specialist" with different equipment (detailed in Fig. 1).

The first ranger (worker), $a_1$, has special equipment for clearing overgrown brush, and the second ranger, $a_2$, has specialized equipment for detecting snares, e.g., a metal detector. Assume 3 states for each patrol area $i$ as "overgrown and snared" ($s = 0$), "clear and snared" ($s = 1$), and "clear and not snared" ($s = 2$). Assume that reward is received only for arms in state $s = 2$, and that snares cannot be cleared from areas with overgrown brush, i.e., $P_{ij}^{02} = 0 \; \forall j \in [M]$. If we assume that each worker is a "true" specialist— so, ranger 1's equipment is ineffective at detecting snares, i.e., $P_{i1}^{12} = 0$, and ranger 2's equipment is ineffective at clearing overgrown brush, i.e., $P_{i2}^{01} = 0$ — then the opti-

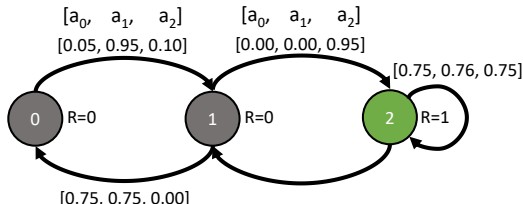

Figure 1: Specialist domain: where specific actions are required in each state to advance to the reward-giving state. Decoupled indices lead to sub-optimal policies, whereas adjusted indices perform well.

mal policy is for ranger 1 to act on the arm in state "overgrown and snared" and ranger 2 to act on the arm in state "clear and snared". However, the fully decoupled index computation for each ranger $j$ would reason about restricted MDPs that only have passive action and ranger type $j$ available. So when computing, e.g., the index for ranger 1 in $s = 0$, the restricted MDP would have 0 probability of reaching state "clear and not snared", since it does not include ranger 2 in its restricted MDP. This would correspond to an MDP that always gives 0 reward, and thus would artificially force the index for ranger 1 to be 0, despite ranger 1 being the optimal action for $s = 0$.

To address this, we define a new index notion that accounts for such inter-action effects. The key idea is that, when computing the index for a given worker, we will consider actions of *all other workers in future time steps.* So in our poaching example, the new index value for ranger 1 in $s = 0$ will *increase* compared to its decoupled index value, because the new index will take into account the value of ranger 2's actions when the system progresses to $s = 1$ in the future. Note that the methods we build generalize to any number of workers $M$. However, the manner in which we incorporate the actions of other workers must be done carefully, We propose an approach and provide theoretical results explaining why. Finally, we give the full algorithm for computing the new indices.

**New index notion**: For a given arm, to account for the inter-worker action effects, we define the new index for an action $j$ as the minimum charge that makes an intervention by $j$ on that arm as valuable as *any* other worker $j'$ in the combined MDP, with $M + 1$ actions. That is, we seek the minimum charge for action $j$ that makes us indifferent between taking action $j$ and *not* taking action $j$, a multi-worker extension Whittle's index notion. To capture this, we define an augmented reward function $R_{\boldsymbol{\lambda}}^{\dagger}(s, j) = R(s) - \lambda_j c_j$. Let $\boldsymbol{\lambda}$ is the vector of $\{\lambda_j\}_{j \in [M]}$ charges. We define this **expanded MDP** as $\mathcal{M}_{\boldsymbol{\lambda}}^{\dagger}$ and the corresponding value function as $V_{\boldsymbol{\lambda}}^{\dagger}$. We now find adjusted index $\lambda_{j, \boldsymbol{\lambda}_{-j}}^{adj,*}$ using the following expression:

$$\min_{j' \in [M] \setminus \{j\}} \arg \min_{\lambda_j} \{\lambda_j \colon V_{\boldsymbol{\lambda}_{-j}}^{\dagger}(s, \lambda_j, j) = V_{\boldsymbol{\lambda}_{-j}}^{\dagger}(s, \lambda_j, j')\} \tag{9}$$

where $\boldsymbol{\lambda}_{-j}$ is a vector of fixed charges for all $j' \neq j$, and the outer min over $j'$ simply captures the specific action $j'$ that the optimal planner is indifferent to taking over action $j$ at the new index value. Note, this is the natural extension of the decoupled two-action index definition, Eq. (5), which defines the index as the charge on $j$ that makes the planner indifferent between acting and, the only other option, being passive. Our new *adjusted index algorithm* is given in Alg. 1.

We use a binary search procedure to compute the adjusted indices since $V_{\boldsymbol{\lambda}_{-j}}^{\dagger}(s, \lambda_j, j)$ is convex in $\lambda_j$. The most important consideration of the adjusted index computation is how to set the charges $\lambda_{j'}$ of the other action types $j'$ when computing the index for action $j$. We show that a reasonable

---
**Algorithm 1** Adjusted Index Computation
---
**Input**: An arm: MDP $\mathcal{M}^{\dagger}$, costs $c_j$, state $s$, and indices $\lambda_j^*(s)$.

1: **for** $j = 1$ to $M$ **do**
2:    $\boldsymbol{\lambda}_j = \lambda_j^*(s)$ {init $\boldsymbol{\lambda}$}
3: **for** $j = 1$ to $M$ **do**
4:    Compute $\lambda_{j,\boldsymbol{\lambda}_{-j}}^{adj,*}(s)$ {via binary search on Eq. 9}
5: **return** $\lambda_{j,\boldsymbol{\lambda}_{-j}}^{adj,*}(s)$ for all workers $j \in [M]$

---

choice for $\lambda_{j'}$ is the Whittle Indices $\lambda_{j'}^*(s)$ which were pre-computed using Alg. 3. The intuition is that $\lambda_{j'}^*(s)$ provides a *lower bound* on how valuable the given action $j'$ is, since it was computed against no-action in the restricted two-action MDP. In Observation 1 and Theorem 2, we describe the problem's structure to motivate these choices.

The following observation explicitly connects decoupled indices and adjusted indices.

**Observation 1.** *For each worker $j$, when $\boldsymbol{\lambda}_{-j} \to \infty$, i.e., $\lambda_{j'} \to \infty \;\; \forall j' \neq j$, then the following holds: $\lambda_{j,\boldsymbol{\lambda}_{-j}}^{adj,*} \to \lambda_j^*$.*

This can be seen by considering the rewards $R_{\boldsymbol{\lambda}}^{\dagger}(s, j') = R(s) - \lambda_{j'} c_{j'}$ for taking action $j'$ in any state $s$. As the charge $\lambda_{j'} \to \infty$, $R_{\boldsymbol{\lambda}}^{\dagger}(s, j') \to -\infty$, making it undesirable to take action $j'$ in the optimal policy. Thus, the optimal policy would only consider actions $\{0, j\}$, which reduces to the restricted MDP of the decoupled index computation.

Next we analyze a potential naive choice for $\boldsymbol{\lambda}_{-j}$ when computing the indices for each $j$, namely, $\boldsymbol{\lambda}_{-j} = 0$. Though it may seem a natural heuristic, this corresponds to planning *without considering the costs of other actions*, which we show below can lead to arbitrarily low values of the indices, which subsequently can lead to poorly performing policies.

**Theorem 2.** *As $\lambda_{j'} \to 0 \;\; \forall j' \neq j$, $\lambda_j^{adj,*}$ will monotonically decrease, if (1) $V_{\lambda_{j'}}^{\dagger}(s, \lambda_j, j') \geq V_{\lambda_{j'}}^{\dagger}(s, \lambda_j, 0)$ for $0 \leq \lambda_{j'} \leq \epsilon$ and (2) if the average cost of worker $j'$ under the optimal policy starting with action $j'$ is greater than the average cost of worker $j'$ under the optimal policy starting with action $j$.*

Thm. 2 (proof in Appendix B) confirms that, although setting $\lambda_{j'} = 0$ for all $j'$ may seem like a natural option, in many cases it will artificially reduce the index value for action $j$. This is because $\lambda_{j'} = 0$ corresponds to planning as if action $j'$ comes with *no charge*. Naturally then, as we try to determine the *non-zero* charge $\lambda_j$ we are willing to pay for action $j$, i.e., the index of action $j$, *we will be less willing to pay higher charges, since there are free actions $j'$*. Note that conditions (1) and (2) of the above proof are not restrictive. The first is a common epsilon-neighborhood condition, which requires that value functions do not change in arbitrarily non-smooth ways with $\lambda$ values near 0. The second requires that a policy's accumulated costs of action $j'$ are greater when starting with action $j'$, than starting from any other action— this is same as assuming that the MDPs do not have arbitrarily long mixing times. That is to say that Thm. 2 applies to a wide range of problems that we care about.

The key question then is: what are reasonable values of charges for other actions $\boldsymbol{\lambda}_{-j}$, when computing the index for action $j$? We propose that a good choice is to set each $\lambda_{j'} \in \boldsymbol{\lambda}_{-j}$ to its corresponding decoupled index value for the current state, i.e., $\lambda_{j'}^*(s)$. The reason relies on the following key idea: we know that at charge $\lambda_{j'}^*(s)$, the optimal policy is indifferent between choosing that action $j'$ and the passive action, at least when $j'$ is the only action available. Now, assume we are computing the new adjusted index for action $j$, when combined in planning with the aforementioned action $j'$ at charge $\lambda_{j'}^*(s)$. Since the charge for $j'$ is already set at a level that makes the planner indifferent between $j'$ and being passive, if adding $j'$ to the planning space with $j$ does not provide any additional benefit over the passive action, *then the new adjusted index for $j$ will be the same as the decoupled index for $j$, which only planned with $j$ and the passive action*. This avoids the undesirable effect of getting artificially reduced indices due to under-charging for other actions $j'$, i.e., Thm. 2. The ideas follow similarly for whether the adjusted index for $j$ should increase or decrease relative to its decoupled index value. I.e., if *higher* reward can be achieved when planning with $j$ and $j'$ together compared to planning with either action alone, as in the specialist anti-poaching example

then we will become *more willing to pay a charge* $\lambda_j$ now to help reach states where the action $j'$ will let us achieve that higher reward. On the other hand, if $j'$ dominates $j$ in terms of intervention effect, then even at a reasonable charge for $j'$, we will be less willing to pay for action $j$ when both options are available, and so the adjusted index will decrease. We give our new *adjusted index algorithm* in Alg. 1, and provide experimental results demonstrating its effectiveness.

### 4.3 Allocation Algorithm

We provide a method called *Balanced Allocation* (Alg. 2) to tackle the problem of allocating intervention tasks to each worker in a balanced way. At each time step, given the current states of all the arms $\{s_i^t\}_{i\in[N]}$, Alg. 2 creates an ordered list $\sigma$ among workers based on their highest Whittle Indices $\max_i \lambda_{ij}(s_i^t)$. It then allocates the best possible (in terms of Whittle Indices) available arm to each worker according to the order $\sigma$ in a round-robin way (allocate one arm to a worker and move on to the next worker until the stopping criterion is met). Note that this satisfies the constraint that the same arm cannot be allocated to more than one worker. In situations where the best possible available arm leads to the budget violation $B$, an attempt is made to allocate the next best. This process is repeated until there are no more arms left to be allocated. If no available arms could be allocated to a worker $j$ because of budget violation, then worker $j$ is removed from the future round-robin allocations and are allocated all the arms in their bundle $D_j$. Thus, the budget constraints are always satisfied. Moreover, in the simple setting, when costs and transition probabilities of all workers are equal, this heuristic obtain optimal reward and perfect fairness.

---

**Algorithm 2** Balanced Allocation

---

**Input**: Current states of each arm $\{s_i\}_{i\in[N]}$, index values for each arm-worker $(i,j)$ pair $\lambda_{ij}(s_i)$, costs $\{c_{ij}\}$, budget $B$, fairness threshold $\epsilon = c_{max}$.
**Output**: balanced allocation $\{D_j\}_{j\in[M]}$ where $D_j \subseteq [N]$, $D_j \cap D_{j'} = \emptyset \ \forall j,j' \in [M]$.
 1: Initiate allocation $D_j \leftarrow \emptyset$ for all $j \in [M]$
 2: Let $L \leftarrow \{1,\ldots,N\}$ be the set of all unallocated arms
 3: **while** true **do**
 4:     Let $\tau_j$ be the ordering over $\lambda_{ij}$ values from highest to lowest: $\lambda[\tau_j[1]][j] \geq \ldots \geq \lambda[\tau_j[N]][j] \geq 0$
 5:     Let $\sigma$ be the ordering over workers based on their highest indices: $\lambda[\tau_1[1]][1] \geq \lambda[\tau_2[1])][2]$ and so on
 6:     **for** $j = 1$ to $M$ **do**
 7:         **if** $\tau_{\sigma_j} \cap L \neq \emptyset$ **then**
 8:             $x \leftarrow \text{top}(\tau_j) \cap L$
 9:             **while** $c_{x\sigma_j} + \sum_{h\in D_{\sigma_j}} c_{h\sigma_j} > B$ **do**
10:                 $\tau_{\sigma_j} \leftarrow \tau_{\sigma_j} \setminus \{x\}$
11:                 **if** $\tau_{\sigma_j} \cap L = \emptyset$ **then**
12:                     break
13:                 **else**
14:                     $x \leftarrow \text{top}(\tau_{\sigma_j}) \cap L$
15:         **if** $\tau_{\sigma_j} \cap L \neq \emptyset$ **then**
16:             $D_{\sigma_j} \leftarrow D_{\sigma_j} \cup \{x\}$;    $L \leftarrow L \setminus \{x\}$;    $\tau_{\sigma_j} \leftarrow \tau_{\sigma_j} \setminus \{x\}$
17: **return** $\{D_j\}_{j\in[M]}$

---

**Theorem 3.** *When all workers are homogeneous (same costs and transition probabilities on arms after intervention) and satisfy indexability, then our framework outputs the optimal policy while being exactly fair to the workers.*

*Proof sketch.* The proof consists of two components: (1) optimality, which can be proved using Corollary 1 (Whittle Indices for homogeneous workers are the same), and the fact that the same costs lead to considering all workers from the same pool of actions, and (2) perfect fairness, using the fact that, when costs are equal, Step 3 of our algorithm divides the arms among workers in a way such that the difference between the number of allocations between two workers differs by at most 1 (see complete proof in Appendix D).

## 5 Empirical Evaluation

We evaluate our framework on three domains, namely **constant unitary costs**, **ordered workers**, and **specialist domain**, each highlighting various challenging dimensions of the MWRMAB problem

275 (detailed in Appendix C). In the first domain, the cost associated with all worker-arm pairs is the
276 same, but transition probabilities differ; the main challenge is in finding optimal assignments, though
277 fairness is still considered. In the second domain, there exists an ordering among the workers such
278 that the highest (or lowest) ranked worker has the highest (or lowest) probability of transitioning any
279 arm to "good" state; which makes balancing optimal assignments with *fair* assignments challenging.
280 The final domain highlights the need to consider inter-action effects via Step 2.

281 We run experiments by varying the number of arms for each domain. For the first and third domains
282 that consider unit costs, we use $B = 4$ budget per worker, and for the second domain where costs are
283 in the range $[1, 10]$, we use budget $B = 18$. We ran all the experiments on Apple M1 with 3.2 GHz
284 Processor and 16 GB RAM. We evaluate the average reward per arm over a fixed time horizon of
285 100 steps and averaged over 50 epochs with random or fixed transition probabilities that follow the
286 characteristics of each domain.

287 **Baselines** We compare our approach, **CWI+BA** (Combined Whittle Index with Balanced Alloca-
288 tion), against:

289 • **PWI+BA** (Per arm-worker Whittle Index with Balanced Allocation) that combines Steps 1 and 3
290    of our approach, skipping Step 2 (adjusted index algorithm)

291 • **CWI+GA** (Combined arm-worker Whittle Index with Greedy Allocation) that combines Steps
292    1 and 2 and, instead of Step 3 (balanced allocation), the highest values of indices are used for
293    allocating arms to workers while ensuring budget constraint per timestep

294 • **Hawkins** [2003] solves a discounted version of Eq. 2 without the fairness constraint, to compute
295    values of $\lambda_j$, then solves a knapsack over $\lambda_j$-adjusted Q-values

296 • **OPT** computes optimal solutions by running value iteration over the combinatorially-sized exact
297    problem (1) without The fairness constraint.

298 • **OPT-fair** follows OPT, but adds the fairness constraints. These optimal algorithms are exponential
299    in the number of arms, states, and workers, and thus, could only be executed on small instances.

300 • **Random** takes random actions $j \in [M] \cup \{0\}$ on every arm while maintaining budget feasibility
301    for every worker at each timestep

302 **Results** Figure 2 shows that reward obtained using our framework (CWI+BA) is comparable to that
303 of the reward maximizing baselines (Hawkins and OPT) across all the domains. We observe at most
304 $18.95\%$ reduction in reward compared to OPT, where the highest reduction occurs for ordered workers
305 in Fig. 2(b). In terms of fairness, Figs. 2(a) and (c) show that CWI+BA achieves fair allocation among
306 workers at all timesteps. In Figure 2(b) CWI+BA achieves fair allocation in almost all timesteps. The
307 fraction of timesteps where fairness is attained by CWI+BA is significantly higher than Hawkins and
308 OPT. In fact, Fig 2(b) also shows that Hawkins obtains *unfair* solutions at every timesteps (0 fairness)
309 when N=5 and B=18, and, when N=10 and N=15, Hawkins is fair only 0.41 and 0.67 fractions of
310 the time, respectively. **Thus, compared to reward maximizing baselines (Hawkins and OPT),**
311 **CWI+BA achieves the highest fairness**. We also compare against two versions of our solution
312 approach, namely, PWI+BA and CWI+GA. We observe that PWI+BA accumulates marginally lower
313 reward while CWI+GA performs poorly in terms of fairness, hence asserting the importance of using
314 CWI+BA for the MWRAMB problem.

315 Fig 3 shows that **CWI+BA is significantly faster than OPT-fair** (the optimal MWRMAB solution),
316 with an execution time improvement of $33\%$, $78\%$ and $83\%$ for the three domains, respectively,
317 when N=5. Moreover, for instances with N=10 onwards, both OPT and OPT-fair ran out of memory
318 because the execution of the optimal algorithms required exponentially larger memory. However, we
319 observe that CWI+BA scales well even for $N = 10$ and $N = 15$ and runs within a few seconds, on
320 an average.

321 Fig. 4 further demonstrates that our **CWI+BA scales well** and consistently outputs fair solution for
322 higher values of $N$ and $B$. On larger instances, with $N \in \{50, 100, 150\}$, our approach achieves up
323 to $374.92\%$ improvement in fairness with only $6.06\%$ reduction in reward, when compared against
324 the reward-maximizing solution Hawkins [2003].

325 **In summary, CWI+BA is fairer than reward-maximizing algorithms (Hawkins and OPT) and**
326 **much faster and scalable compared to the optimal fair solution (OPT fair), while accumulating**

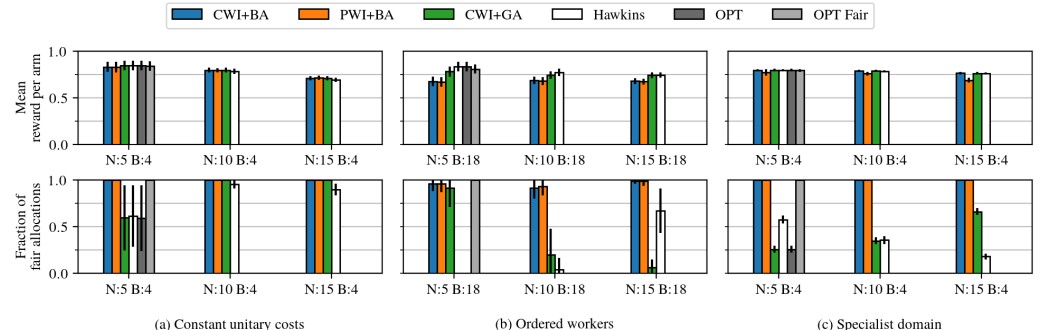

Figure 2: **Mean reward** (top row) and **fraction of time steps with fair allocation** (bottom row) for $N = 5, 10, 15$ arms. CWI+BA (blue) achieves highest fraction of fair allocations than Hawkins (white) algorithm while **attaining almost similar reward as the reward-maximizing baselines**.

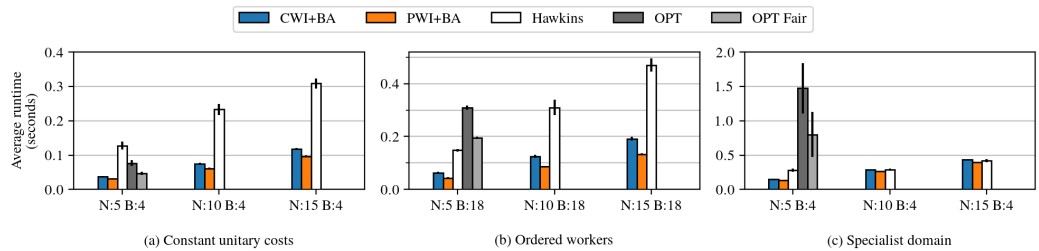

Figure 3: **Execution time** averaged over 50 epochs for $N = 5, 10, 15$. For a fixed time horizon of 100 steps, CWI+BA run faster than Hawkins (white), OPT (dark gray), and OPT fair (light gray) for all instances in each of the three domains evaluated.

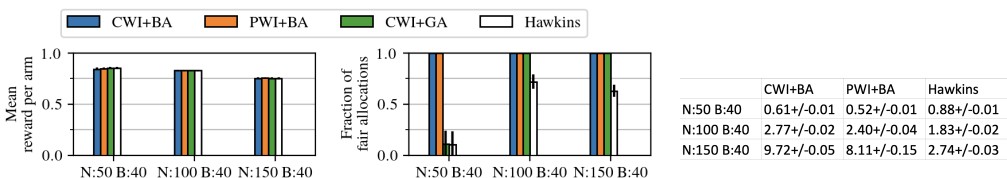

|           | CWI+BA      | PWI+BA      | Hawkins     |
|-----------|-------------|-------------|-------------|
| N:50 B:40 | 0.61+/-0.01 | 0.52+/-0.01 | 0.88+/-0.01 |
| N:100 B:40| 2.77+/-0.02 | 2.40+/-0.04 | 1.83+/-0.02 |
| N:150 B:40| 9.72+/-0.05 | 8.11+/-0.15 | 2.74+/-0.03 |

Figure 4: The plot shows **mean reward** (left), **fairness** (middle), and **run time** (right) for $N = 50, 100, 150$ arms on **constant unitary costs** domain. CWI+GA scales well for larger instances, and even for N=150 arms, the average runtime is 10 seconds.

**reward comparable to Hawkins and OPT across all domains.** Therefore, CWI+BA is shown to be a fair and efficient solution for the MWRMAB problem.

## 6   Conclusion

We are the first to introduce multi-worker restless multi-armed bandit (MWRMAB) problem with worker-centric fairness. Our approach provides a scalable solution for the computationally hard MWRMAB problem. On comparing our approach against the (non-scalable) optimal fair policy on smaller instances, we find almost similar reward and fairness.

Our problem formulation provides a more general model for the intervention planning problem capturing heterogeneity of intervention resources, and thus it is useful to appropriately model real-world domains such as anti-poaching patrolling and machine maintenance, where the interventions are provided by a human workforce.

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
