# OpenReview forum: "Fairness for Workers Who Pull the Arms: An Index Based Policy for Allocation of Restless Bandit Tasks"
_NeurIPS.cc/2022/Conference — NeurIPS 2022 Submitted_

### Official Review · Reviewer_TAkH · 2022-07-11

**Rating:** 7
**Confidence:** 3
**Soundness:** 3 good
**Presentation:** 3 good
**Contribution:** 2 fair

**Summary:**

This paper considers constrained resource allocation tasks framed as restless bandit problems. In contrast to prior work in this space which assumes that all of the resources (i.e., healthcare workers, mechanics, etc.) to be allocated are interchangeable, the authors: (a) consider *heterogeneous* workers with individual/varied costs, budgets, and intervention effects; and (b) strive to optimally allocate arms to workers while ensuring the disparity in costs incurred by any two workers does not exceed a given threshold, $\epsilon$. To do this, the authors define an augmented action space, such that each worker is represented as a possible active action, with a corresponding worker-arm-specific cost function and transition matrix. They then decompose the problem and compute Whittle indices for each worker-arm combination. They use these Whittle indices to compute an adjusted index that reflects the minimum charge required to make the agent indifferent between an active action provided by worker $j$ and one provided by any other worker $j^\prime$. They use the adjusted indices to inform budget- and fairness-constraint satisfying round-robin allocation of arms to workers.

**Questions:**

1. How is fair allocation defined/operationalized in the empirical results?
2. Could the knapsack portion of the Hawkins approach be modified to encode the type of cost-disparity threshold fairness considered here?

**Limitations:**

The authors acknowledge that incorporating fairness among workers (i.e., with respect to the number of arms each worker is allocated) will, in general, reduce expected total reward relative to a fairness-agnostic (but otherwise equivalent) approach. An additional limitation is that worker-specific transition matrices are assumed to be known by the agent.

**Strengths And Weaknesses:**

*Strengths*:
- The paper is well-written and the extension to the RMAB setting that the authors propose (i.e., actions with arm-specific heterogenous costs and effectiveness) is well-motivated by many practical domains, such as healthcare, manufacturing, and wildlife conservation. The technical work required to extend the problem setting in this way is significant since one must now account for how workers with different (arm/state-specific) expertise may interact (and influence arms' transitions) over time.

*Weaknesses*:
- Arms' states and worker-specific transition matrices are assumed to be observable. While it is conventional to assume each arm's (worker-agnostic) transition matrix is known by the decision-maker in the planning setting, the assumption that this additional level of information is available should be explicitly stated and justified by the availability/"learnability" of this type of information within the domain(s) of interest. This will be particularly important in settings where there is heteroskedastic uncertainty about workers' intervention effects.

---

> ### Author Response · Authors · 2022-08-02
> **Response to Reviewer TAkH**
>
> We thank the reviewer for their positive and encouraging feedback. Here are the answers to the questions.
> 1. We consider a solution to be fair when it ensures that the difference between costs incurred between any two agents is at most c_max, at each time step.
> 2. Yes, a fairness constraint can be added to the Hawkins approach; however, Hawkin’s computation becomes even slower by adding a fairness constraint.

---

### Official Review · Reviewer_QfKj · 2022-07-12

**Rating:** 3
**Confidence:** 4
**Soundness:** 1 poor
**Presentation:** 2 fair
**Contribution:** 2 fair

**Summary:**

This paper considers a multi-worker version of the restless multi-armed bandit problem where the intervention on a particular arm can be carried out by one of M possible workers. The workers have possibly different costs for intervening on different arms and the goal is to design an optimal allocation policy that satisfies budget constraint for each worker and also fair.

The authors devise a heuristic balanced allocation rule for the problem. The main idea behind the algorithm is to decompose the original problem into several related subproblems. The experiments show that the proposed heuristic scales with the number of agents without compromising too much on the objective.



**Questions:**

1. The original optimization problem (1 and 2) has the fairness constraint. However, this constraint is absent in the subproblem (3). How do you guarantee that solving each subproblem will automatically generate a fair solution?

2. Also I am not sure when solving each subproblem is sufficient to solve the original problem~(2). Do you have any result showing that you can obtain a solution of the original problem by solving the $N\times M$ subproblems?

3. I don't understand why theorem 1 is useful. It seems to me this theorem is only used to derive Corollary 1 for the case of identical cost and probability transition function. But doesn't that directly follow from the nature of the optimization problem (3) by substituting identical cost and probability transition function?

4. Theorem 3 states that the algorithm balanced allocation works only when the workers are homogeneous. But I thought the whole point of introducing the general framework was to handle workers with heterogeneous costs. Do you have any results for the setting with heterogeneous costs?

5. The balanced allocation takes fairness threshold $\epsilon$ as input. But as far as I can understand, the algorithm never explicitly uses this argument. It would be nice if you could clarify which step of the algorithm uses this parameter? Also regarding this, do you have any guarantee for the fairness guarantee obtained by the balanced allocation algorithm?

**Limitations:**

This paper doesn't have any negative societal impact.

**Strengths And Weaknesses:**

Strengths:
- I think the multi-worker extension of the restless multi-armed bandit problem is an interesting direction and certainly more practical.
- The empirical section is interesting as the authors could show that the proposed approximation algorithm runs fast for a large number of agents ($N$ is of the order of hundreds) and the performance doesn't deteriorate much.

Weaknesses:
- The main weakness of the paper is the lack of proper theoretical guarantees. First, it is not clear if you can decompose the original problem~(2) into $N \times M$ subproblems because of the additional fairness constraint. Second, the main result also shows that the proposed algorithm works only for homogeneous workers (theorem 3) and it's not clear if the balanced allocation works when the costs are heterogeneous.
- Even if I assume that the problem is decomposable the authors don't bound fairness violation of the final solution. Note that fairness is a global constraint and the individual subproblems don't have this constraint. So it is not trivial to combine the solutions of these subproblems and ensure a desired level of fairness.
- For the adjusted index algorithm it is not clear what is the right choice of the variables $\lambda_{-j}$ for each $j$. The authors justified that choosing $\lambda_{-j} = 0$ is not optimal, and chose these values based on Whittle indices. But I am not sure that should be optimal either because of heterogeneous costs and global constraints. Maybe you can solve for a fixed point of this system and use the fixed point as the adjusted indices?

Some other comments:
- Give definition of indexability of arms.
- At the end of section 3, the authors highlight two challenges for MWRMAB model. It might be a good idea to illustrate these challenges with an example.

---

> ### Author Response · Authors · 2022-08-02
> **Response to Reviewer QfKj**
>
> We thank the reviewer for valuable comments. Please find below our responses to the questions:
> 1. Our proposed heuristic provides a systematic solution to the original problem by using a three-step solution. The first two steps address the multi-worker RMAB problem (without fairness constraint), and the third step tackles the fairness problem. Solving the subproblems is a tool used to find the "intervention benefit" of each arm-worker pairing, that must be paired with a fair allocation scheme (e.g., balanced allocation), to solve the original problem~(2)
> 2. The solutions of the subproblems (steps 1 and 2) may not solve the original problem-(2), and need to be paired with a fair allocation procedure.
> 3. After removing the fairness constraint and considering time-averaged budget constraints, the Lagrangian objective can be written in terms of summation over N and summation over M, and thus, allowing it to be decomposed into NXM subproblems.
> 4. Theorem 1 provides a closed form for computing an index for one worker relative to the index of some other worker, when the transition probabilities are equal. This allows for computational speedup. That is, the planner only needs to carry out an expensive index computation for a single worker, then the planner can use the computed index to compute indices for the other workers in O(1) time. This is useful when any subset of workers have the same transition probability functions, but different costs. Note that the traditional RMAB framework assumes there is only a single worker with a single transition function, so Theorem 1 covers a natural multi-worker extension of traditional RMABs in which workers have the same probability functions (e.g., intervention effectiveness on each arm), but have different costs (e.g., are located in different areas and have different travel times to reach each arm.)
> 5. We do not have any guarantees for heterogeneous costs. However, we demonstrate empirically that our proposed algorithm achieves excellent fairness while sacrificing very little in solution quality across various simulated domains.
> As shown in Theorem 4, the balanced allocation algorithm is guaranteed to be optimally fair when the costs are all equal. An interesting future direction is to find more general instances where “balanced allocation” guarantees fairness.

---

### Official Review · Reviewer_H4ZJ · 2022-07-12

**Rating:** 6
**Confidence:** 4
**Soundness:** 2 fair
**Presentation:** 3 good
**Contribution:** 3 good

**Summary:**

The authors study the problem of multi-worker restless bandits, that extends the standard restless bandits having different costs and fairness constraints on different actions. The authors propose a scheme that at first decouples the problem and than, using the decoupled solutions, introduce the interaction with other workers. Finally, some experiments comparing the proposed algorithm to the ones present in the literature (which do not include the fairness constraint) are provided.

**Questions:**

It is not clear to me if the solution you provided is solving the original problem or if it is an approximate solution. In the former case, I would like to have a formal statement about that, in the latter case a study on the approximation gap.

Is Theorem 3 only predicating on those instances having that all the workers are of the same type? I would like to understand if in this theorem we are restricting to a specific case or not.

**Limitations:**

I do not foresee limitations and potential negative societal impacts. The paper is focused on the idea to make the work allocation task fair, therefore at the moment i only see positive impacts on society.

**Strengths And Weaknesses:**

The paper is well structured and the definition of the steps taken to get to the solution is helping the reader to understand the idea behind the paper. There are a few ortographic mistakes. The topic is interesting and the solution is well motivated empirically.

I have some doubts about the real formulation of some of the results and the strenght of the theoretical results provided.


Minor:
Line 99: "M=1" I got the idea, but i think that the more formal statement would be c_ij = c_ih=1 \forall j,h \in [M]
Line 130: It is not clear the statement. Is it "by relying on the work by qian et al."? Or is it a derivation from the previous work?
Line 283: Please provide the rationale behind the choice of B)18.
Line 297: The -> the
Figure 4: I suggest to rearrange the subplots and the table, to better explain the results therein.

---

> ### Author Response · Authors · 2022-08-02
> **Response to Reviewer H4ZJ**
>
> We thank the reviewer for their valuable feedback. We first address the minor comments and then the questions listed.
>
> - In line 99, we wanted to point out that the MWRMAB problem reduces to the well-studied RMAB problem when there is only one worker with costs 1 for intervening any arm. For this implication, the suggestion by the reviewer to add “c_ij=c_ih=1” is not enough, it would then require two additional changes: (1) transition probabilities are also the same for all workers and (2) the fairness constraint is omitted.
>
> - Line 130: For step 1 of our proposed heuristic, the algorithm by Qian et al. (also, described in the Appendix) is directly used for computing the Whittle Indices.
>
> - For the experiments, we conducted experiments on a range of N and B values and then added a selected set of plots in the paper. The second domain has heterogeneous costs (ranging between 1-10) and hence B=18 (or any B>15) is a suitable budget to ensure that some workers are allocated at least 2 interventions.
>
> - We arranged the subplots to show the reward, fairness, and time complexity of the three domains. We are happy to rearrange the plots and tables to explain the results better, but it is not clear whether the reviewer’s suggestion is to separate out the table or make some other change.
>
> Answering the questions that the reviewer raised:
>
> 1. The fair worker RMAB problem is a challenging combinatorial problem with no known solution or proven approximation. We develop a heuristic algorithm, guided by theoretical insights on the problem structure, which we demonstrate empirically achieving excellent fairness while marginally sacrificing the solution quality.
>
> 2. Yes, Theorem 3 assumes the case that all workers have the same transition probability functions and cost functions. We show that if there is a fairness constraint added to the RMAB problem with multiple homogeneous workers, our algorithm gives the same solution as OPT-fair (the optimal solution for Problem-2 satisfying fairness constraint).

---

> > ### Comment · Reviewer_H4ZJ · 2022-08-08
> > **Comments after rebuttal**
> >
> > I thank the authors for the time spent addressing my concerns.
> >
> > I think that my opinion on the paper did not change substantially.
> >
> > About my comment about the rearrangement of Figure 4, I meant that the figures and the table should be provided into two different graphical elements (for instance, using \minipage) so that they have two separate captions.

---

### Meta-Review · Area_Chair_37Wv · 2022-08-30

**Recommendation:** Reject
**Confidence:** Certain

**Metareview:**

This paper looks at the restless MAB (RMAB) problem through the lens of fairness, an increasingly-active area over the last few years.  This works generalizations to fairness (over the arms that are pulled) but spread out across multiple workers who are pulling the arms.  This is a well-motivated setting, and one that might see application in various mobile-health-related scenarios that are popular in this space, and where the ML community has a presence.  Still, reviewers raised many questions about theoretical bounds for cases under the purview of the settings investigated.  We would appreciate a stronger rebuttal and/or stronger edits to a camera-ready or next submission for this work.

**Award:**

No

---

### Decision · Program_Chairs · 2022-09-14

Reject